# Exploring the Relationships among Brand Experience, Perceived Product Quality, Hedonic Value, Utilitarian Value, and Brand Loyalty in Unmanned Coffee Shops during the COVID-19 Pandemic

**Jun-Ho Bae [1] and Hyeon-Mo Jeon [2,\*]**

1    Department of Hotel & Tourism Administration, Halla University, Wonju 26404, Korea
2    Department of Hotel, Tourism, and Foodservice Management, Dongguk University-Gyeongju, Gyeongju 38066, Korea
\*    Correspondence: jhm010@dongguk.ac.kr; Tel.: +82-10-6275-4010

**Abstract:** This study aims to find the antecedents that enhance consumer value and brand loyalty to unmanned coffee shops (UCS) that provide unmanned services during the COVID-19 pandemic. The analysis developed and tested a series of hypotheses based on data collected from 463 customers who had visited UCS in South Korea. The influence of brand experience and perceived product quality on customers' hedonic and utilitarian values were examined, and the utilitarian values were found to have a significant effect on brand loyalty. This result signifies the importance of brand experience and perceived product quality in inducing consumers' perceptions of value and brand loyalty in the context of unmanned services. The study's design and results differ from those of previous brand experience studies on manned stores in the food service industry. Therefore, this study contributes to hospitality literature by applying brand experience theory, which has been applied to research on human and unmanned services. In addition, it makes an important contribution by presenting practical implications for the sustainable management of the food service industry during the COVID-19 era.

**Keywords:** unmanned coffee shops; brand experience; perceived product quality; hedonic value; utilitarian value; brand loyalty





## 1. Introduction

With the health crisis triggered by COVID-19, most countries have faced financial downturns due to the restrictions imposed to curb the spread of the pandemic [1]. This unexpected crisis has had an enormous effect on the tourism industry. Consequently, the food service industry has been hit the hardest [2]. Consumers have started avoiding crowded places and limiting restaurant visits due to social distancing guidelines for COVID-19 prevention and control [3]. Nevertheless, in Starbucks stores in South Korea, employees were infected with COVID-19 through customers, which in turn spread rapidly to other customers [4]. This is evidence of the risk of face-to-face services given the spread of the virus that occurred in a face-to-face service between employees and customers. For these reasons, there has been an accelerated transition toward contactless shopping due to the influence of COVID-19, and the public is getting used to digital shopping in the pandemic era [5]. Digital retail has revolutionized the number of people who choose to shop face-to-face or online, including unstaffed stores [5]. Unmanned stores offer self-service checkpoints and safety for non-present employees, as well as dynamic pricing systems [5]. Even in the food service industry, the number of unmanned smart restaurants providing services to customers without direct intervention by staff is rising, such as Amazon Go in the U.S., McDonald's Future Restaurant 2.0, unmanned vending machines across Japan, Alibaba's unmanned restaurant, and JD JOY's smart restaurant in China [6]. In South Korea, human

baristas have recently disappeared from coffee shops, and 24-h unmanned coffee shops (UCS) equipped with robots and coffee vending machines are emerging. Currently, the survey results show that about 820 unmanned coffee shops are operating nationwide, and sales more than doubled in 2021 compared to 2020 [7]. This phenomenon is attributed to easy operation in a small space without labor costs and customer preferences for contactless consumption [8]. As such, unmanned services are being implemented as a risk-reduction approach and as a potential panacea for mitigating the risk of the spread of COVID-19 [9].

Therefore, understanding the new patterns of consumer consumption behavior for unmanned services in the coffee business sector is important to meet consumer needs and expectations quickly [6]. UCS are expected to further expand as a new form of coffee shop business that will allow management efficiency and sustainability by reducing labor costs as well as consumer safety from COVID-19 or other infectious diseases. Despite their importance, sufficient empirical research on fully unmanned store services, except for human services in the coffee shop business sector, has not been conducted. Looking at the research on unmanned services in the coffee shop business sector, Sung and Jeon [10] confirmed the consumers' intention to accept robot barista services in coffee shops with human baristas. In addition, Kim and Ryu [11] and Hwang et al. [12] verified consumer behavior regarding take-out robot barista coffee shops. Although previous studies [10–12] have shown consumer behavioral studies of several robotic barista service situations, consumer experiences with fully store-type UCS are still not being investigated. However, the number of unmanned coffee shops that have completely excluded human baristas due to COVID-19 has rapidly increased in recent years. To reflect this trend in the coffee shop business, this study was conducted on unmanned store-type coffee shops (such as Starbucks) without human baristas rather than unmanned take-out coffee shops. This finding differs from the results of previous studies [10–12].

Therefore, this study focuses on the relationships among customer experience, product quality, value, and brand loyalty in UCS. Experience can originate from many factors, but it can also shape or change the consumer's perception of the brand [13]. Consumers' brand experience is not automatically created but is formed by consumer perception, emotion, and behavior while interacting with marketing factors such as products, services, and brand space [14]. When a brand provides a more positive experience to consumers, pleasant and familiar emotions are induced, and positive emotions and emotional consumption values toward the brand are formed in consumers' minds [13,15]. In addition, consumers' consumption of a brand not only allows them to acquire utilitarian value for products and services, but also build loyalty to the brand through various experiences related to brand purchases [16]. Until now, brand experience theory in the hospitality industry has mainly been applied to studies on manned hotels or restaurants [13,17–19], but this study intends to apply it to the increasingly expanding UCS.

The biggest difference from previous studies is related to the application of brand experience. In addition, this study applies perceived product quality as a factor that increases consumers' hedonic and utilitarian value of the brand experience. The importance of food quality in dining experiences has been emphasized according to various characteristics [20]. When consumers perceive the quality of food service brands as high, they achieve consumption goals and develop hedonic emotions toward the brand [18]. A previous study [21] argues that food quality in organic restaurants is essential for the perceived monetary value. Therefore, perceived product quality is an important determinant of consumer value [21]. The perceived hedonic and utilitarian value of a consumer's experience in a restaurant is effective for enhancing the intention to revisit and has been identified as an important variable that induces customer loyalty [13,22]. Therefore, it is necessary to gain an in-depth understanding of the predictors that affect consumers' brand loyalty to pursue safety in the pandemic and meet the needs of digital consumers who crave new experiences. Therefore, this study proposes the perceived product quality as a positive antecedent to hedonic and utilitarian value and brand loyalty among UCS consumers alongside the four sub-factors

of brand experience proposed by Brakus et al. [23]: namely, sensory, affective, behavioral, and intellectual.

As such, this study aims to identify the factors that affect consumer value and brand loyalty for UCS use by presenting the brand experience and perceived product quality based on previous studies. This research design is different from previous brand experience studies [13,17–19] that have focused on manned stores in the food service industry. The results of the analysis suggest important factors that can predict consumer behavior for fully unmanned food service stores in the future and will be conducive to the hospitality literature. In addition, this study presents practical implications for the establishment of sustainable management strategies for food service brands in the era of the COVID-19 pandemic and thereafter.

## 2. Literature Review and Hypotheses

### 2.1. Brand Experience

Brand experience has been extensively examined and discussed as a concept in marketing and branding literature and has received much attention from scholars and practitioners [24]. Schmitt [14] argued that marketers should go beyond simply marketing product features and instead aim to provide consumers with a great experience [25]. Therefore, experiential marketing, which emphasizes consumers' experiences in the consumption process of branded products and services, has emerged over traditional marketing, which emphasizes the features and benefits of products and services [14]. Brand experience is likely to influence consumers' behavioral responses [23–27]. Therefore, entrepreneurs prioritize selling special brand experiences because doing so attracts more sales and repeated customer engagement [12,28,29].

Brand experience is triggered by a variety of stimuli that arise when customers interact with a brand's service organization, products, and employees [14,27]. In this context, Schmitt [14] first proposed the concept of strategic experience modules (SEMs) from a multidimensional perspective of cognitive, sensory, affective, visual, and social perspectives. However, Schmitt's [14] SEMs consist of conceptual elements that appeared in the consumers' basic information processing process about brands, so there is a limitation in that the classification of each element is unclear [23]. In addition, it has a limitation in that it only focuses on the behavioral response of consumers from the functional attributes of the brand, but does not examine the overall experience induced by the brand itself [30]. Considering this inherent complexity, Brakus et al. [23] subsequently defined the brand experience as "specific sensations, feelings, cognitions, and behavioral responses elicited by stimuli associated with the brand that are part of the brand's design and identity, packaging, communications, and environments". They conceptualized and empirically established brand experience as a multidimensional construct consisting of four dimensions: sensory, affective, behavioral, and intellectual. Jain, Aagja, and Bagdare [31] described customer experience as "cognitive, emotional, sensorial, and behavioral responses formed throughout the decision-making process and conclusion chain, which includes a series of interactions integrated with people, objects, processes, and environment". Among the four dimensions of brand experience proposed by Brakus et al. [23], the sensory brand experience refers to the response of sensory organs, such as vision, hearing, and touch induced by the brand. This is a consumer's aesthetic response, which also creates an overall impression of a brand, such as brand personality [32]. An effective brand experience is an emotional response that consumers receive from a brand. The brand's emotional experience can arise from contact and interaction between the consumer and the brand, and can lead to an emotional bond between them [33]. A behavioral brand experience is the behavioral and physical response that consumers experience through the brand and stems from the motivation to express their lifestyle and self-concept. This is mainly found during interactions beyond sensory and emotional experiences with other consumers. It involves sharing thoughts with friends and family and writing online reviews [26]. An intellectual brand experience is a consumer's cognitive response to a brand. It promotes creative thinking by stimulating

curiosity and interest in the brand [34]. Thus, an intelligent brand experience ensures that consumers do not feel bored [29]. A brand experience consisting of these four dimensions is considered as the experience of consumers interacting with the brand, and when these dimensions harmonize with each other, the overall brand experience is complete [23,35]. This concept of four-dimensional brand experience has been verified in various product and service settings [36], such as tourist attractions [37], consumer events [38], personal care products [39], airlines [40], coffee houses [12,29,41], and restaurants [35].

### 2.2. Perceived Product Quality

Consumers compare perceptions of product quality before and after using a product [42]. Quality is generally defined as a consumer's cognitive response to a service encounter [43]. Zeithaml [44] explained that the perceived quality differs from the actual quality due to higher levels of abstraction involving the consumer's holistic overall assessment of the experience. Perceived quality is a consumer's subjective evaluation of the overall superiority or excellence of products and services provided by a company. High perceived quality signifies that consumers perceive a brand as differentiated and superior through experience [35]. The literature on hospitality studies establishes perceived food quality as a key determinant of restaurant quality [25,45–47]. According to Chen et al. [47], customers prioritize food quality over other factors such as price, value, convenience, and cleanliness. Previous studies emphasizing the importance of food quality in the dining experience [46,48–51] evaluated food quality according to various characteristics. Kivela et al. [48] defined taste, menu variety, and nutrition as the main attributes of food quality, whereas Ha and Jang [50] established taste, portion, menu variety, and healthy options as elements of perceived food quality. Ryu et al. [46] proposed taste, nutrition, menu variety, freshness, food aroma, and visual appeal as the food quality elements for upscale restaurants. Han et al. [51] explained that the key quality aspects of in-flight meals and beverages are related to the key characteristics of food and beverages, such as taste, nutrition, and freshness of ingredients. Yu and Fang [49] proposed that taste, purity, and quality of coffee are the product quality elements of a coffee shop. The measurement of perceived quality can be found in the general premise that the consumer forms a perceived quality assessment by recalling various aspects of the overall experience [52]. Several restaurant studies have confirmed a significant relationship between brand experience and perceived quality. Ding and Tseng [18] confirmed that brand experience has a positive effect on perceived quality in a study of ice cream, coffee, and fast-food brands. Jeon and Yoo [35] verified that the brand experience of a grocerant, a combination of groceries and restaurants, improves overall quality, including food. As such, previous studies have identified brand experience as an important antecedent that improves the quality perceived by consumers in the context of restaurants. Therefore, we propose brand experience as an important predictor of customers' perceived value of the brand in the context of UCS. Based on the results of previous studies, the following hypothesis was proposed:

**Hypothesis 1 (H1).** *Brand experience has a significant positive effect on perceived product quality.*

### 2.3. Value

Zeithaml [44] conceptualized value as "a consumer's overall assessment of the utility of a product (or service) based on the perception of what it received and what it gave." [24] Babin, Darden, and Griffin [53] suggest that consumption activities can produce both hedonic and utilitarian results; consumers evaluate whether utilitarian motivation is achieved while consuming a brand [54]. Utilitarian motivation and emotion appear as emotional motivations to benefit from the product, that is, hedonic motivation [55].

Therefore, researchers have classified values into utilitarian (or functional) and hedonic (or empirical) values [56]. Holbrook [57] established a value system by dividing the value of the consumption experience into an extrinsic–intrinsic value dimension and an active–reactive value dimension. Extrinsic value refers to a utilitarian aspect of the

experience, which is obtained through product acquisition and exchange, while intrinsic value refers to a value such as fun or pleasure generated by simply liking the action itself rather than the acquisition of the object. Utilitarian value is defined as an overall assessment of functional benefits that incorporates four aspects: price reduction, service, time-saving, and product selection [58]. As an efficient and functional value, utilitarian value is rational and in line with the objectives [59]. Hedonic value based on emotional motivation refers to the excitement or pleasure that occurs when consumers are immersed in a shopping environment and are looking around the product, which means that the emotional and irrational aspects are more powerful than when consumers acquire something through shopping [59]. For example, the aesthetics of the environment of a grocery store or restaurant means that the purchasing process is enjoyable or that you can escape boredom by experiencing a happy purchasing experience [60]. Clearly, utilitarian and hedonic values are considered fundamental to understanding consumers' assessments of the consumption experience because they maintain a underlying presence throughout the phenomenon of consumption [53,61]. Several studies have demonstrated that hedonic and utilitarian values are important for consumers to experience brands. Cachero-Martínez and Vázquez-Casielles [62] demonstrated that a brand experience that incorporates sensory, intellectual, and social experiences has a positive influence on hedonic value. Yoo et al. [13] confirmed that brand experience, hedonic value, and utilitarian value have a significant influence on consumers who experience grocerants. Rodrigues and Brandão [24] confirmed that the IKEA brand experience increases hedonic and utilitarian values. Particularly, the study identified that brand experience strengthens hedonic rather than utilitarian values. Coelho, Bairrada, and Matos Coelho [63] confirmed that brand experience enhances monetary value, and according to Ahn and Back's study [64], the experience of cruise brand influences the functional value. Ding and Tseng [18] state that the experience of a food service brand forms customers' hedonic feelings. Based on the results of previous studies, the following hypotheses were proposed:

**Hypothesis 2 (H2).** *Brand experience has a significant positive effect on hedonic value.*

**Hypothesis 3 (H3).** *Brand experience has a significant positive effect on utilitarian value.*

Oliver [65] implied that perceived value is a higher concept that includes quality. Some empirical results consistent with Oliver's [65] conceptual reasoning have shown that quality is a key determinant of value perception. Smith and Colgate [66] confirmed that products that create experiences, feelings, and emotions appropriate to consumers have a significant relationship with utilitarian and hedonic values. Cachero-Martínez and Vázquez-Casielles [62] confirmed that pragmatic experiences such as price, product, and promotion are important factors that increase utilitarian value. Jeon and Yoo [34] stated that the perceived quality of a grocerant (food and service) has a positive effect on the perception of economic and hedonic values. Yu and Fang [49] confirmed that Starbucks' coffee quality is valuable for consumption. Ding and Tseng [18] confirm that the perceived quality of a food service brand is an antecedent that increases customers' hedonic feelings. Based on the results of previous studies, the following hypotheses were proposed:

**Hypothesis 4 (H4).** *Perceived product quality has a significant positive effect on hedonic value.*

**Hypothesis 5 (H5).** *Perceived product quality has a significant positive effect on utilitarian value.*

*2.4. Brand Loyalty*

Strong brands play an important role in building brand loyalty, particularly in competitive business environments [25,67]. Loyalty can be defined as a customer's attachment to or feelings toward a brand and/or company [68–70]. Oliver [71] defined loyalty to a brand as a deeply held commitment to continuously repurchase or patronize products and

services, despite various efforts and circumstances that induce conversion behavior. With high brand loyalty, consumers have a higher tendency to recommend the brand to others while also establishing a lower tendency to purchase an alternative brand [72]. Creating loyal customers is a major goal of service marketers and often represents a fundamental component of a company's long-term competitive strategy [73]. Customer brand loyalty is critical to sustainability in the food service industry [74]. Loyal customers are a prerequisite for a strong customer base and higher market share [74–76]. Existing research on brand loyalty has focused on post-consumption responses such as attitudinal and behavioral loyalty [25,77,78]. Specifically, behavioral loyalty is centered on the repurchase process that attracts customers and encourages loyalty to a particular brand [79], whereas attitude loyalty is defined as "a customer's tendency toward a brand as a result of psychological stimulation" [70].

Consumers tend to be loyal to a brand, expand their experience, and continue to achieve hedonistic and utilitarian goals [18,54,80]. In this context, a handful of researchers have demonstrated an important relationship between hedonic and utilitarian values and brand loyalty in the business and hospitality sectors. Jones, Reynolds, and Arnold [81] argued that, in retail shopping situations, the greater the consumer's perception of hedonic and utilitarian shopping value, the higher the loyalty. Ryu et al. [61] stated that the hedonic and utilitarian values perceived by fast-food restaurant customers have a positive effect on revisits and recommendation intention. Yoo et al. [13] confirmed that the hedonic and utilitarian values perceived by customers who have experienced a grocerant are precedent factors that influence the intention to revisit. In particular, it was identified that hedonic values have a greater influence than utilitarian values. Based on the results of previous studies, the following hypotheses were proposed:

**Hypothesis 6 (H6).** *Hedonic value has a significant positive effect on brand loyalty.*

**Hypothesis 7 (H7).** *Utilitarian value has a significant positive effect on brand loyalty.*

All the hypotheses are in the theoretical model, as depicted in Figure 1.

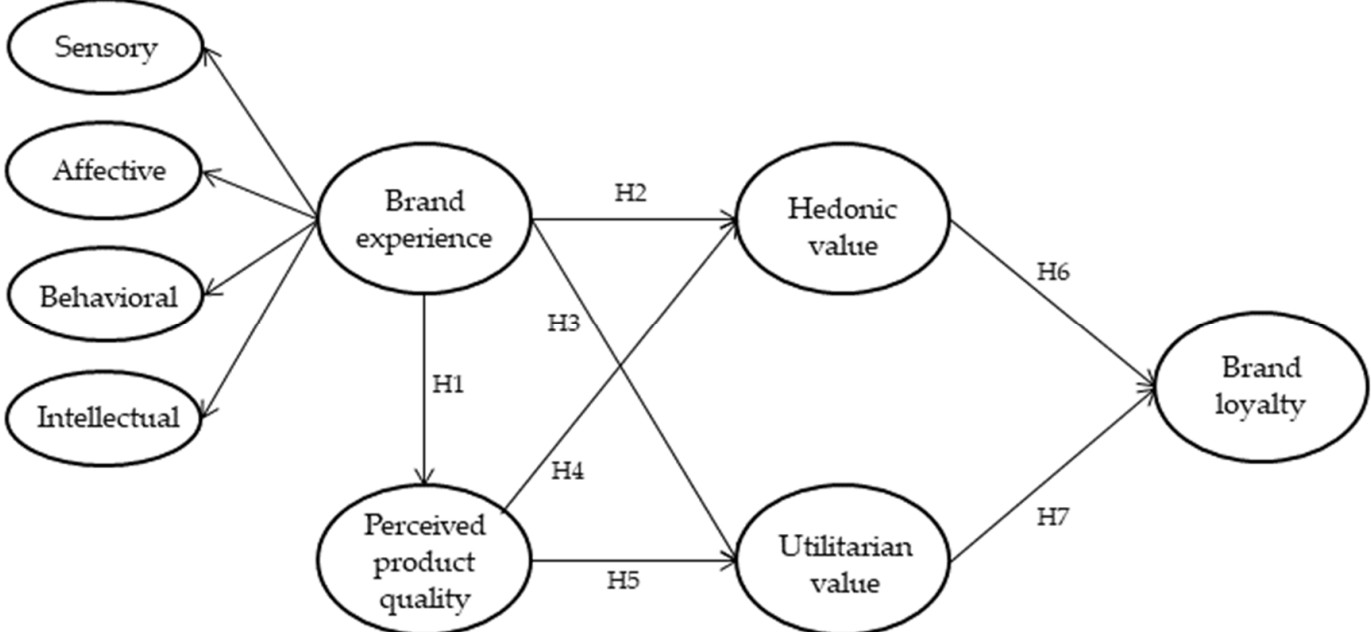

**Figure 1.** Proposed conceptual model.

## 3. Methodology

### 3.1. Research Instrument

The questionnaire was composed of items based on previous research and was modified to suit the context of the UCS (see Appendix A). Before the survey, a pilot test was conducted to determine the participants' level of understanding and to select the UCS brand to be surveyed. A total of eight factors were used for this study. First, the 4 sub-dimensions of brand experience consisted of 12 items (three each) derived from Brakus et al. [23], Choi et al. [17], and Hwang et al. [12]. Perceived product quality was measured using four items from Yu and Fang [49] and Ryu et al. [46]. Yoo et al. [13] cited four hedonic and three utilitarian items [13]. Finally, brand loyalty was measured using four items developed by Choi et al. [19] and Kim et al. [4]. All items were measured using a five-point Likert-type scale ranging from one (strongly disagree) to five (strongly agree).

### 3.2. Pilot Test

A pilot test was conducted to evaluate the 12-item questionnaire using convenience sampling through an online SNS. Participants were asked two screening questions about the definition of UCS and whether they had visited the UCS before being selected for the sample. We then performed an exploratory factor analysis (EFA) to verify the dimensionality of the 12 items of brand experience with the collected data. Data were collected from 100 respondents. The sample size was considered sufficient because it met the criterion of a minimum subject-to-item ratio of 5:1 [82]. The most visited UCS by 100 respondents was Coffeebanhada (29%) followed by Café AI (12%) and B;eatbox (10%). The remaining UCS utilization rate was less than 10% each.

The EFA revealed that the factor loading of all items was less than 0.4, and the overall construct reliability was less than 0.7 [83], which satisfy the acceptable values. This process produced a 4-factor solution containing 12 items, which together accounted for 68.305% of the total variables [84]. The first factor included three sensory components ($\alpha = 0.685$), the second factor included three affective components ($\alpha = 0.730$), the third factor included three behavioral components ($\alpha = 0.777$), and the fourth factor included three intellectual components ($\alpha = 0.769$).

### 3.3. Sampling and Data Collection

UCS handle the entire process of ordering, payment, manufacturing, notification, and pickup by a barista without an employee. Usually, employees visit once a day for 30 min to an hour to clean the store and replenish raw materials. There are largely two types of UCS, a 'robot barista type' using robot arms and a 'vending machine type' with a built-in coffee bean grinder and extractor. The 'robot barista type' is a method that receives information directly from kiosks, and when a customer orders, a robot that receives a signal immediately makes the coffee through the machine and hands it to the customer [10]. For comparison with manned coffee shops, this study targets a store-type UCS that allows customers to stay in the store for a prolonged time while consuming coffee.

The participants of the survey were selected from among adults aged 20 years or older who mainly go to coffee shops and residents of Korea who have visited UCS within the last three months. Through the pilot test, Coffeebanhada, Café AI, and B;eat box, which are identified as UCS most frequently used by coffee consumers, were selected as the brands to be examined. The users of these three UCS are representative of all UCS' users. In particular, Coffeebanhada is a franchise coffee shop brand that operates more than 600 manned stores and 33 unmanned stores nationwide. It has the largest number of stores among unmanned coffee shops, making it a brand that consumers can easily use [85].

Data were collected online through ENTRUST, a specialized research company, from 1 May to 14 May 2022. Two screening questions were administered before the respondents were invited to participate in the interviews. Have you visited one or more of "Coffeebanhada, Café AI, B;eat box" within the last three months? Did you know that "Coffeebanhada, Café AI, B;eat box" is a UCS? If either answer was no, the survey was closed. Of the

6573 people who responded to the survey, 505 answered "yes" to the two questions. After explaining the purpose of this study to the 505 respondents, the research institute obtained informed consent and conducted the survey. The questions were presented to all respondents in the same order, and 505 questionnaires were collected during the survey period. After dropping 42 questionnaires with identical answers for all metrics, 463 valid questionnaires were coded into the final dataset, indicating an effective return rate of 91.7%. Table 1 presents the sample profiles.

**Table 1.** Respondents' profiles.

| Demographic Characteristics | | Frequency | Percentage |
|---|---|---|---|
| Gender | Male | 208 | 44.9 |
| | Female | 255 | 55.1 |
| Age | 20–29 years | 132 | 28.5 |
| | 30–39 years | 108 | 23.3 |
| | 40–49 years | 96 | 20.7 |
| | 50–59 years | 82 | 17.7 |
| | Above 60 years | 45 | 9.7 |
| Educational level | High school | 64 | 13.8 |
| | 2-year university | 27 | 5.8 |
| | 4-year university | 284 | 61.3 |
| | Graduate school | 88 | 19.0 |
| Monthly income | Below USD 2000 | 70 | 15.1 |
| | USD 2000–2999 | 103 | 22.2 |
| | USD 3000–3999 | 83 | 17.9 |
| | USD 4000–4999 | 72 | 15.6 |
| | USD 5000–5999 | 50 | 10.8 |
| | Above USD 6000 | 85 | 18.4 |
| Occupation | Office workers | 194 | 41.9 |
| | Professional job | 88 | 19.0 |
| | Self-employed | 59 | 12.7 |
| | Others | 59 | 12.7 |
| | Student | 37 | 8.0 |
| | Sales and Service | 26 | 5.9 |

### 3.4. Analytical Methods

A two-step approach [86] was used to empirically analyze the research hypothesis.

The first step was to check the fit of the measurement model and evaluate the reliability, convergent validity, and discriminant validity using confirmatory factor analysis (CFA). CFA, a high-order factor analysis, was conducted as a second-order factor. Second, structural equation modeling (SEM) was performed to verify the hypothetical relationships among the five components proposed in the conceptual model.

## 4. Results

### 4.1. Measurement Model

CFA was performed to confirm the appropriateness of the measurement structure, and seven common model fit measurements were used [87]: $\chi^2/df$ (<3), GFI (>0.9), RMSEA (<0.08), RMR (<0.08), NFI (>0.9), IFI (>0.9), and CFI (>0.9). The CFA results showed that the model had adequate fit statistics ($\chi^2 = 314.586$, df = 132, $\chi^2/df = 2.383$, RMR = 0.018, GFI = 0.936, NFI = 0.939, IFI = 0.964, CFI = 0.964, RMSEA = 0.055) (see Table 2). In addition, as shown in Table 2, the composite construction reliability (CCR) of the components was higher than 0.70 [87], indicating that all components of the model had internal consistency. All average variance extraction (AVE) estimates were above the recommended threshold of 0.05 [87], indicating that convergent validity was supported. CCR and AVE were calculated using standard loading and variance of error for each item [87]. Finally, the AVE square root of each variable was compared to the correlation coefficient between any pair of

constructs. The AVE square root of each variable was higher than the corresponding correlation coefficient, thereby confirming the discriminant validity [88] (see Table 3).

**Table 2.** Measurement model assessment.

| Variables and Item | SL | CCR | AVE |
|---|---|---|---|
| Brand experience (BE) (α = 0.870) | | | |
| Sensory | 0.772 | 0.941 | 0.799 |
| Affective | 0.846 | | |
| Behavioral | 0.764 | | |
| Intellectual | 0.792 | | |
| Perceived product quality (PQ) (α = 0.800) | | | |
| PQ1 | 0.752 | 0.894 | 0.678 |
| PQ2 | 0.691 | | |
| PQ3 | 0.738 | | |
| PQ4 | 0.715 | | |
| Hedonic value (HV) (α = 0.791) | | | |
| HV1 | 0.729 | 0.868 | 0.622 |
| HV2 | 0.720 | | |
| HV3 | 0.692 | | |
| HV4 | 0.671 | | |
| Utilitarian value (UV) (α = 0.746) | | | |
| UV1 | 0.670 | 0.847 | 0.650 |
| UV2 | 0.702 | | |
| UV3 | 0.743 | | |
| Brand loyalty (BL) (α = 0.839) | | | |
| BL1 | 0.747 | 0.899 | 0.690 |
| BL2 | 0.741 | | |
| BL3 | 0.704 | | |
| BL4 | 0.832 | | |

Note: SL, standard loading; CCR, composite construct reliability; AVE, average variance extracted.

**Table 3.** Correlations of analysis between the variables.

| Variable | 1 | 2 | 3 | 4 | 5 |
|---|---|---|---|---|---|
| 1. BE | 0.894 | | | | |
| 2. PQ | 0.676 | 0.823 | | | |
| 3. HV | 0.744 | 0.658 | 0.789 | | |
| 4. UV | 0.735 | 0.629 | 0.724 | 0.806 | |
| 5. BL | 0.718 | 0.733 | 0.655 | 0.671 | 0.831 |
| Mean | 3.701 | 3.813 | 3.819 | 3.908 | 3.787 |
| S.D. | 0.552 | 0.571 | 0.604 | 0.600 | 0.638 |

Note: Diagonal elements show the square root of AVE. Below the diagonal is the corresponding correlation coefficient. All correlation coefficients are significant at the 0.001 level.

*4.2. Common Method Bias*

As this was a cross-sectional study with a single source of data, common method bias (CMB) may exist. Therefore, we performed Harman's single-factor test to determine whether there was CMB. EFA was performed by loading all items of the variable into a single factor without rotation. EFA was conducted by injecting the 27 remaining measurement items of brand experience, perceived product quality, hedonic value, utilitarian value, and brand loyalty. As a result of the analysis, the variance of a single factor was explained as 44.5%. Therefore, a criterion of less than 50% was met [89,90]. Thus, it was confirmed that there were no CMB problems.

*4.3. Structural Model*

SEM was performed to identify the seven hypotheses, and the results are presented in Table 4. The model fit indices were $\chi^2$ = 288.618, df = 129, *p* = 0.000, $\chi^2/\text{df}$ = 2.237, RMR = 0.017, GFI = 0.942, NFI = 0.944, IFI = 0.968, and CFI = 0.968, RMSEA = 0.052, thereby meeting the standard assessment criteria. H1 was supported because brand experience positively and significantly influences perceived product quality ($\beta$ = 0.774, t = 13.446, *p* = 0.000). H2 was supported because brand experience positively and significantly influences hedonic value ($\beta$ = 0.839, t = 10.747, *p* = 0.000). H3 was supported because brand experience positively and significantly influences utilitarian value ($\beta$ = 0.779, t = 9.926, *p* = 0.000). H4 was supported because perceived product quality positively and significantly influences hedonic value ($\beta$ = 0.141, t = 2.206, *p* = 0.027). H5 was supported because perceived product quality positively and significantly influences utilitarian value ($\beta$ = 0.238, t = 3.304, *p* = 0.000). H6 was rejected because the hedonic value did not significantly influence brand loyalty ($\beta$ = 0.086, t = 0.413, *p* = 0.679). H7 was supported because utilitarian value positively and significantly influences brand loyalty ($\beta$ = 0.933, t = 4.252, *p* = 0.000).

**Table 4.** Results of the structural model analysis.

| | Hypotheses | $\beta$ | t-Value | *p*-Value | Decision |
|---|---|---|---|---|---|
| H1 | BE → PQ | 0.774 | 13.446 ** | 0.000 | supported |
| H2 | BE → HV | 0.839 | 10.747 ** | 0.000 | supported |
| H3 | BE → UV | 0.779 | 9.926 ** | 0.000 | supported |
| H4 | PQ → HV | 0.141 | 2.206 * | 0.027 | supported |
| H5 | PQ → UV | 0.238 | 3.304 ** | 0.000 | supported |
| H6 | HV → BL | 0.086 | 0.413 | 0.679 | rejected |
| H7 | UV → BL | 0.933 | 4.252 ** | 0.000 | supported |

Note: * *p* < 0.05, ** *p* < 0.01.

## 5. Discussion

This study aims to verify consumers' value perception of UCS by applying perceived product quality to the brand experience proposed by Brakus et al. [23]. Additionally, the study was conducted to verify the influencing relationship between brand loyalty as an outcome variable and the consumers' perceived value. This model incorporates various explanatory variables, such as sensory, affective, behavioral, intellectual, perceived product quality, hedonic value, and utilitarian value, as determinants of brand loyalty to UCS.

The results of the data analysis confirmed that brand experience increases perceived product quality. It was confirmed that brand experience, which integrates sensory, emotional, behavioral, and intellectual experiences, is an important antecedent that positively evaluates product quality. This result is consistent with previous studies [18,35] that have investigated the relationship between brand experience and quality in the food service business. In addition, the study demonstrates that brand experience and perceived product quality are positive determinants of customer value perception of UCS. In particular, the brand experience was found to have the greatest influence on the hedonic and utilitarian value of UCS, proving that the customer's experience is very important. This is consistent with the results of previous studies [14,18,24] that brand experience establishes customer value. Schmitt [14] states that consumers can obtain more value than just information from brands if they engage in favorable and unique experiences with products and brands. In addition, this study supports Rodrigues and Brandão [24], who argued that brand experience is important in enhancing customers' hedonic and utilitarian values. This is consistent with the study by Ding and Tseng [18], who stated that a holistic experience in food service brands induces hedonic emotions. This means that the more customers that experience the sensory, affective, behavioral, and intellectual senses when visiting UCS, the more positively they perceive hedonic values such as pleasure, interest, and joy, as well as utilitarian values, such as convenience and practicality. What is particularly noteworthy in this result is that customers feel hedonic value more than utilitarian value while

experiencing a UCS. This signifies that UCS are not only more convenient, practical, and economically positioned for customers than manned coffee shops, but also more interesting and enjoyable.

Perceived product quality also strengthens customers' hedonic and utilitarian values. This implies that, while it is important to enhance value through positive experience, customers' perceived hedonic and utilitarian values improve when favorable perceptions of product quality are included. This partially supports the study of Yu and Fang [49], who proved that the product quality of manned coffee shops is an important factor in determining value. They argued that the taste, purity, and overall quality of coffee provide value to customers. Ryu et al. [46] verified the customers' value of food quality as a single dimension of practicality, but this study differs from theirs in that it verified value by dividing it into hedonic and utilitarian values. Therefore, the results of this analysis will provide a more practical customer perception of value and will be an important basis for researching the customers' value of product quality in the food service business.

This study proved that utilitarian value formed through the brand experience of UCS is effective in increasing customer brand loyalty. In addition, unlike face-to-face service situations, it was confirmed that the hedonic value of customers in UCS has no influence on brand loyalty. This partially differs from the studies of Ryu et al. [61] and Yoo et al. [13], which proved that hedonic and utilitarian values are important variables for increasing customer revisit and recommendation intentions in the context of restaurants. Brand loyalty is expected to predict continuous visits and positive word-of-mouth interactions from customers in the future. The results of this study are significant in that the customers had value and loyalty to the UCS brand through experience and product quality in the context of UCS. The results of this analysis have the following theoretical and practical implications.

## 6. Implications

### 6.1. Theoretical Implications

One of the most important theoretical implications is relevant to the brand-experience theory used in this study. Brand experience is a vital differentiation tool for obtaining the value and loyalty of restaurant customers [74]. This study, in consideration of public safety in the COVID-19 era, intended to identify the antecedents that affect the value perception and brand loyalty of UCS that emerged as a form of food service business. We applied the brand experience theory of Brakus et al. [23], which has received attention as a theory that can strengthen brand loyalty and create brand value with the antecedent. However, although many previous studies have investigated the importance of applying this theory, we believe that there is a limit to applying this theory to the food service industry. Therefore, a model was designed to add qualitative factors that are important characteristics of food service products. This model is different from those of previous studies [13,17,27,74] that applied the brand experience theory in the food service industry. These studies applied the empirical theory of Brakus et al. [23] to products of food service brands but overlooked product quality. Our study goes one step further by including the perceived product quality as a variable. Therefore, this study identifies the structural relationship between these variables by applying Brakus et al.'s [23] brand experience and perceived product quality as antecedents of customers' hedonic and utilitarian value perceptions and brand loyalty to UCS for the first time, and confirms the suitability of the model.

The results confirm that brand experience theory, which has been applied to studies based on experience in staffed stores or face-to-face services in the food service industry [13,17,27,74], can be applied even to non-face-to-face service situations. Finally, it was verified through this study that the role of brand experience and perceived product quality, which integrates sensory, affective, behavioral, and intellectual responses, are very important for increasing consumer value perception and brand loyalty to UCS. This study adds to the current literature on the topic of brand experience theory and applies this theory in the context of UCS. These research designs and results also support and expand the existing literature on the experiences of food service brands. In addition, this study is expected

to be useful for sustainable restaurant management at a time when social discussions are needed regarding digital transformation, which has accelerated due to COVID-19.

### 6.2. Practical Implications

Our research highlights the importance of the UCS brand experience's four dimensions and perceived product quality in creating hedonic and utilitarian values across the full range of consumer and brand relationships. It is very important to make a strong impression on the customers' five senses. A variety of coffee menus that can stimulate the tastes of customers should be developed, and customized services must be provided in consideration of their preferences. Additionally, coffee cups, sleeves, and goods should be manufactured to arouse customers' visual interest. Given that a UCS does not have human workers and only robots and vending machines, it is not easy to appeal to customers' emotions.

To consumers' emotions, the UCS should create a sensuous and impressive interior and provide a comfortable and cozy atmosphere through internal lighting, music, and spatial arrangement. In addition, to induce vitality in consumers' daily behavior and stimulate intellectual curiosity, new experiences based on 4th Industrial Revolution technology, such as AI robots, chatbots, and big data, should be constantly provided to consumers. The experience provided based on the digital system, from pre-arrival order to post-arrival stay in the store, provides convenience and interest to customers.

Saved labor costs in unmanned operations should be reflected in product prices so that customers can enjoy products at lower costs. We have confirmed that utilitarian value influenced by brand experience is the most effective in shaping brand loyalty. Therefore, the UCS must strengthen its utilitarian value by providing customers with convenient and reasonable product prices. In addition, coffee products should be prepared in variety, and their quality should be the same as that of manned coffee shops. Moreover, all ingredients, including coffee beans, should be managed in a fresh state, and the hygiene of the coffee-making devices should be maintained. Product quality management strengthens customers' utilitarian and hedonic values.

In contrast, hedonic values did not significantly affect the brand loyalty of UCS. It is predicted that customers have a desire for pleasure gained from interactions with employees in manned coffee shops as compared to UCS where there are no such interactions. To overcome this problem, it is necessary to use anthropomorphic humanoid robots or chatbots that communicate with customers. If an individual develops an application for the UCS brand and opens a chatbot service to provide customers with various information, such as the brand introduction and coffee menu description, the lack of interaction with customers will be resolved. Customers who visit will experience a different kind of pleasure through simple conversations (such as greetings and simple orders) with the humanoid robot.

Even in the current or post-COVID-19 era, coffee consumers who are accustomed to social distancing are expected to continue using UCS. Therefore, when non-face-to-face services are established, the UCS becomes a sustainable management model for the food service business.

### 7. Limitations and Future Research

This study has a few limitations. First, since the data were collected only from South Korea, there are limitations in generalizing the results of the study; since each country has a different acceptance level and coffee culture when it comes to the 4th Industrial Revolution technology, it may not be appropriate to apply the results of this study to other countries. Second, because online surveys can cause selection bias [91], future studies should use other types of data collection methods, such as face-to-face surveys with customers, to reduce bias and increase response rates. Third, participants were limited to USC customers. In the future, an analysis of the comparison between customers' brand experience in UCS and manned coffee shops can help understand the needs of coffee consumers in the pandemic era more clearly.

**Author Contributions:** J.-H.B. and H.-M.J. conceived and designed the experiments; H.-M.J. performed the experiments and analyzed the data; J.-H.B. and H.-M.J. wrote the paper. All authors have read and agreed to the published version of the manuscript.

**Funding:** This research received no external funding.

**Institutional Review Board Statement:** Not applicable.

**Informed Consent Statement:** Not applicable.

**Data Availability Statement:** The data presented in this study are available on request from the corresponding author.

**Conflicts of Interest:** The authors declare no conflict of interest.

## Appendix A

**Table A1.** Measurement items for study.

| Measurement Items | |
|---|---|
| **Sensory (SE)** | |
| SE1 | This UCS brand makes a strong impression on my visual sense or other senses. |
| SE2 | I find this UCS brand interesting in a sensory way. |
| SE3 | This UCS brand appeals to my senses. |
| **Affective (AF)** | |
| AF1 | This UCS brand induces feelings and sentiments. |
| AF2 | I have strong emotions for this UCS brand. |
| AF3 | This UCS brand is an emotional brand. |
| **Behavioral (BH)** | |
| BH1 | When I have coffee at this UCS brand, I feel active and energetic. |
| BH2 | After I have coffee at this UCS brand, I think I can work on tasks more effectively. |
| BH3 | When I am reminded of this UCS brand, I feel lively. |
| **Intellectual (IN)** | |
| IN1 | I engage in a lot of thinking when I encounter this UCS brand. |
| IN2 | When I think about this coffee brand, I am reminded of how it succeeds with its creative strategies. |
| IN3 | This UCS brand stimulates my curiosity and problem-solving. |
| **Perceived product quality (PQ)** | |
| PQ1 | The coffee and beverage product were delicious. |
| PQ2 | The UCS offered a variety of coffee and beverage items. |
| PQ3 | The UCS offered fresh coffee and beverage. |
| PQ4 | Overall, the product quality of this UCS is high. |
| **Hedonic value (HV)** | |
| HV1 | The use of this UCS makes me feel good. |
| HV2 | The use of this UCS is fun and pleasant. |
| HV3 | The use of this UCS is truly a joy. |
| HV4 | While using this UCS, I feel excitement in searching for a coffee product. |
| **Utilitarian value (UV)** | |
| UV1 | The use of this UCS is convenient. |
| UV2 | The use of this UCS is pragmatic and economical. |
| UV3 | While using this UCS, I can easily find the coffee product I need and want. |
| **Brand loyalty (BL)** | |
| BL1 | I will continue to visit this UCS brand in the future. |
| BL2 | If I need coffee, this UCS brand would be my preferred choice. |
| BL3 | I am highly likely to recommend this UCS brand to friends and relatives. |
| BL4 | I will spread positive word-of-mouth about this UCS brand. |

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
