# Peer review of "Exploring the Relationships among Brand Experience, Perceived Product Quality, Hedonic Value, Utilitarian Value, and Brand Loyalty in Unmanned Coffee Shops during the COVID-19 Pandemic"

_sustainability, doi:10.3390/su141811713_

Round 1

Reviewer 1 Report (Previous Reviewer 3)

I found there are severe methodological flows that did not correctly adjust based on comments.

Author Response

Response to Reviewer 1 Comments

* The revised contents of the manuscript are presented in red.

Point 1: I found there are severe methodological flows that did not correctly adjust based on comments.

Response 1: Thank you for your comment. We have revised the manuscript in its entirety from the introduction to the discussion and implications.

We hope that our revised paper adequately meets your requirements and is worthy of being published in your esteemed journal.

Reviewer 2 Report (Previous Reviewer 2)

The authors well-revised the paper based on the reviewers' suggestions.

Author Response

Response to Reviewer 2 Comments

* The revised contents of the manuscript are presented in red.

Point 1: The authors well-revised the paper based on the reviewers' suggestions.

Response 1: Thank you for your comment.

We hope that our revised paper adequately meets your requirements and is worthy of being published in your esteemed journal.

Reviewer 3 Report (New Reviewer)

1.       Background/ Introduction
the innovation of the paper is not enough. The current version of the introduction is composed of 5 paragraphs (around 2 pages). The introduction should preferably be 1.5 pages. Your introduction should be concise and wanting in terms of how it explains what the paper is and what you intend to accomplish. In general, I suggest you state more emphatically and clearly why the topic is important. I feel you need to do a better job of "explaining" the importance of this work to the reader. I am in favor of a succinct but strong introduction. The intro paragraph should be the state of the literature, the second paragraph - what is missing and why it matters, and the third paragraph is how the paper is going to address this gap.

2.       You need to enlarge the literature background and references. Some authors are cited many times in the document and support several affirmations and hypotheses with the same references all over again. Please update references with the latest developments in the sector and include a more varied sources of knowledge about constructs of study. I am including some references that might help, this does not mean you necessary include them all in the manuscript:

-Chua, A. Y. K., & Banerjee, S. (2013). Customer knowledge management via social media: The case of Starbucks. Journal of Knowledge Management, 17(2), 237–249. https://doi.org/10.1108/13673271311315196

-Song, H. J., Wang, J. H., & Han, H. (2019). Effect of image, satisfaction, trust, love, and respect on loyalty formation for name-brand coffee shops. International Journal of Hospitality Management, 79, 50–59. https://doi.org/10.1016/j.ijhm.2018.12.011

-Ibrahim, B. (2021). The Nexus between Social Media Marketing Activities and Brand Loyalty in Hotel Facebook Pages: A Multi-Group Analysis of Hotel Ratings. Tourism: An International Interdisciplinary Journal, 69(2), 228-245.

-Ibrahim, B. 2022. “Social Media Marketing Activities and Brand Loyalty: A Meta-Analysis Examination.” Journal of Promotion Management. 1–31. doi:10.1080/10496491.2021.1955080

3.       sampling technique- Author(s) mentioned that they had used non-probability and convenience sampling techniques. The non-random sample has to be treated with great care, and this weakness should be recognized more explicitly.

4.       Methods-Since the data was collected from one source, the author(s) must explain how they dealt with Common method bias.

5.       Theoretical implications: I would recommend you highlight how your findings help enrich our understanding.

6.       The current version of the discussion section does not sufficiently highlight the contributions of the study. The discussion part is relatively simple. In the part of discussion, the paper simply compares the previous research, and does not discuss how to deepen the theoretical research.

7.       The practical implications are not actionable and speculative at best.

Author Response

Response to Reviewer 3 Comments

* The revised contents of the manuscript are presented in red.

Point 1: 1. Background/ Introduction

the innovation of the paper is not enough. The current version of the introduction is composed of 5 paragraphs (around 2 pages). The introduction should preferably be 1.5 pages. Your introduction should be concise and wanting in terms of how it explains what the paper is and what you intend to accomplish. In general, I suggest you state more emphatically and clearly why the topic is important. I feel you need to do a better job of "explaining" the importance of this work to the reader. I am in favor of a succinct but strong introduction. The intro paragraph should be the state of the literature, the second paragraph - what is missing and why it matters, and the third paragraph is how the paper is going to address this gap.

Response 1: Thank you for your comment. We have modified the introduction to be more concise and powerful as suggested.

Point 2: You need to enlarge the literature background and references. Some authors are cited many times in the document and support several affirmations and hypotheses with the same references all over again. Please update references with the latest developments in the sector and include a more varied sources of knowledge about constructs of study. I am including some references that might help, this does not mean you necessary include them all in the manuscript:

Response 2: We have expanded the literature background and references as suggested.

Point 3: sampling technique- Author(s) mentioned that they had used non-probability and convenience sampling techniques. The non-random sample has to be treated with great care, and this weakness should be recognized more explicitly.

Response 3: We have described the sampling techniques in more detail, as per your comment.

Point 4: Methods-Since the data was collected from one source, the author(s) must explain how they dealt with Common method bias.

Response 4: We have described CMB as per your comment.

Point 5: Theoretical implications: I would recommend you highlight how your findings help enrich our understanding.

Response 5: We modified this section according to your suggestion.

Point 6: The current version of the discussion section does not sufficiently highlight the contributions of the study. The discussion part is relatively simple. In the part of discussion, the paper simply compares the previous research, and does not discuss how to deepen the theoretical research.

Response 6: We have added content in consideration of your comments.

Point 7: The practical implications are not actionable and speculative at best.

Response 7: We have revised the content in consideration of your comments.

We hope that our revised paper adequately meets your requirements and is worthy of being published in your esteemed journal.

Reviewer 4 Report (New Reviewer)

Dear Authors,

ABSTRACT (in short reply to):

- First of all, what is the goal or purpose of this article?

- Specify the results you have achieved,

- Specify how this research differs from others, for example: What is the added value? What is new here?

Introduction:

- To bring the above -mentioned themes more closer to the above mentioned

Article Structure:

- Structuralize according to the recommendations of a scientific journal

Advanced:

- Which brand in that area was the best?

- What are the reasons for choosing the product / brand for consumers?

Author Response

Response to Reviewer 4 Comments

* The revised contents of the manuscript are presented in red.

Point 1: ABSTRACT (in short reply to):

- First of all, what is the goal or purpose of this article?

- Specify the results you have achieved,

- Specify how this research differs from others, for example: What is the added value? What is new here?

Response 1: Thank you for your comment. We have revised the abstract according to your comments.

Point 2: Introduction:

- To bring the above -mentioned themes more closer to the above mentioned

Response 2: We revised and supplemented the introduction as suggested.

Point 3: Article Structure:

- Structuralize according to the recommendations of a scientific journal

Response 3: We have revised the entire manuscript from the introduction to the discussion and implications.

Point 4: Advanced:

- Which brand in that area was the best?

- What are the reasons for choosing the product / brand for consumers?

Response 4: We have described the brands that consumers prefer the most and why they choose them.

We hope that our revised paper adequately meets your requirements and is worthy of being published in your esteemed journal.

Reviewer 5 Report (New Reviewer)

Few suggestions to improve the paper:

1. Line 74 - 77: These statements require references because they reflect/support the research gaps.

2. Line 365 - 366: Justify why the min age of respondents must be 20.

3. Line 367 - 369: The selection of these cafes require supporting references.

4. Line 587 - 588: Suggest the authors to specifically mention other data collection methods that they suggested.

5. Line 391 - 400: Suggest the authors to write the descriptive results in the form table to make it more presentable.

Author Response

Response to Reviewer 5 Comments

* The revised contents of the manuscript are presented in red.

Point 1:  Line 74 - 77: These statements require references because they reflect/support the research gaps.

Response 1: Thank you for your comment. We've added the relevant content as suggested.

Point 2: Line 365 - 366: Justify why the min age of respondents must be 20.

Response 2: We have corrected the relevant sentence as per your comment.

Point 3: Line 367 - 369: The selection of these cafes require supporting references.

Response 3: We have described cafe selection in a pilot test.

Point 4: Line 587 - 588: Suggest the authors to specifically mention other data collection methods that they suggested.

Response 4: We mentioned other data collection methods as suggested.

Point 5: Line 391 - 400: Suggest the authors to write the descriptive results in the form table to make it more presentable.

Response 5: We tabulated Respondents' profiles as suggested.

We hope that our revised paper adequately meets your requirements and is worthy of being published in your esteemed journal.

Round 2

Reviewer 1 Report (Previous Reviewer 3)

I would thank the authors for their hard work.

This manuscript is a resubmission of an earlier submission. The following is a list of the peer review reports and author responses from that submission.

Round 1

Reviewer 1 Report

This study is well written with clear communication and more importantly addresses a very timely appropriate issue. The introduction section provided strong justification for conducting this study by emphasizing the existing gap in prior relevant literature. It leads smoothly the theoretical background based on prior relevant literature. The implications clearly provide what this study revealed and how those findings could possibly contribute to the existing relevant knowledge. I believe that this study can be a significant contribution to future studies for further investigation on relevant topics. 

Reviewer 2 Report

Comments:

Thanks to the author(s) for a well-researched manuscript. address a few minor suggestions.

Review of literature for the concepts in the paper are too weak. Conduct another round of literature review, and provide a lot stronger literature review.

Conclusion: The discussion is a little disappointing and could be written in a much stronger way. Also, there are very little implications, and more specific, poignant recommendations should be provided to the future researchers. This information would give the paper much better finish.

Reviewer 3 Report

1. language editing required

2. in the introduction, the authors explained, “Despite their importance, sufficient empirical research on unmanned services in the food service industry has not been conducted.” However, I am afraid I have to disagree with the authors' claim saying that insufficient empirical studies were conducted in the food service industry in connection with service robots or unmanned services. I just googled quickly and found the empirical research below focusing on the abovementioned topics. The authors should expand the scope to service robot / unmanned service / kiosk / or other similar technologies in the food service industry to improve the quality of the introduction and literature review.

Kim, H. M., & Ryu, K. (2021). Examining image congruence and its consequences in the context of robotic coffee shops. Sustainability13(20), 11413.

Hwang, J., Choe, J. Y. J., Kim, H. M., & Kim, J. J. (2021). Human baristas and robot baristas: How does brand experience affect brand satisfaction, brand attitude, brand attachment, and brand loyalty? International Journal of Hospitality Management99, 103050.

Hwang, J., Choe, J. Y. J., Kim, H. M., & Kim, J. J. (2021). The antecedents and consequences of memorable brand experience: Human baristas versus robot baristas. Journal of Hospitality and Tourism Management48, 561-571.

Jain, N. R. K., Liu-Lastres, B., & Wen, H. (2021). Does robotic service improve restaurant consumer experiences? An application of the value-co-creation framework. Journal of Foodservice Business Research, 1-19.

Mende, M., Scott, M. L., van Doorn, J., Grewal, D., & Shanks, I. (2019). Service robots rising: How humanoid robots influence service experiences and elicit compensatory consumer responses. Journal of Marketing Research56(4), 535-556.

3. This study does not have any theoretical perspective. A theoretical framework should be provided for rigorous scientific research. Why and how are proposed constructs connected – not from previous empirical studies but the theoretical perspective?

4. The current version of the introduction is not convincing and lacks logical flow.

5. please describe the current market size or situation of unmanned coffee shops in Korea in greater detail.

6. for the sample selection, why were people visiting other than four unmanned shops excluded in the study? They, anyway, visited the unmanned shops. Please explain the particular reason for this sample selection in the manuscript. 

7. Possibly, the collected data is severely suffered from selection bias. How do the authors overcome the such issue? Please explain in the manuscript.

8. My calculation of CCR and AVE based on SL in table 1 is entirely different from the value in Table 1. For example, AVE for sensory should be .421, and CCR should be .684. Please re-check and provide the equation used by the author.

Intuitively, among three indicators, two SLs were relatively lower than the threshold (0.7), but AVE is over .54. It does not make sense. Please check again.

9. for Table 2, the diagonal elements could be lower than the given value (if the previous comments are valid). Then, discriminant validity cannot be established.

10. the brand experience was constructed as a second-order factor. Please follow the proper procedure for verifying validity using a second order. 

11. theoretical implications are somewhat repetitive to the previous section. Please revise and strengthen your argument